# TRANSFERABLE RECOGNITION-AWARE IMAGE PROCESSING

## ABSTRACT

Recent progress in image recognition has stimulated the deployment of vision systems (e.g. image search engines) at an unprecedented scale. As a result, visual data are now often consumed not only by humans but also by machines. Meanwhile, existing image processing methods only optimize for better human perception, whereas the resulting images may not be accurately recognized by machines. This can be undesirable, e.g., the images can be improperly handled by search engines or recommendation systems. In this work, we propose simple approaches to improve machine interpretability of processed images: optimizing the recognition loss directly on the image processing neural network or through an intermediate transforming model, a process which we show can also be done in an unsupervised manner. Interestingly, the processing model's ability to enhance the recognition performance can transfer when evaluated on different recognition models, even if they are of different architectures, trained on different object categories or even different recognition tasks. This makes the solutions applicable even when we do not have the knowledge about future downstream recognition models, e.g., if we are to upload the processed images to the Internet. We conduct comprehensive experiments on three image processing tasks with two downstream recognition tasks, and confirm our method brings substantial accuracy improvement on both the same recognition model and when transferring to a different one, with minimal or no loss in the image processing quality.

## 1 INTRODUCTION

Unlike in image recognition where a neural network maps an image to a semantic label, a neural network used for image processing maps an input image to an output image with some desired properties. Examples include image super-resolution (Dong et al., 2014), denoising (Xie et al., 2012), deblurring (Eigen et al., 2013), colorization (Zhang et al., 2016) and style transfer (Gatys et al., 2015). The goal of such systems is to produce images of high perceptual quality to a human

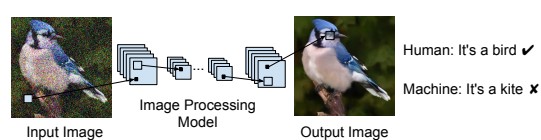

Figure 1: Image processing has been used to generate images that look good for human, but not machines. In this work we study the problem of making processed images more recognizable by machines.

observer. For example, in image denoising, we aim to remove noise in the signal that is not useful to an observer and restore the image to its original "clean" form. Metrics like PSNR and SSIM (Wang et al., 2004) are often used (Dong et al., 2014; Tong et al., 2017) to approximate human-perceived similarity between the processed images with the original images, and direct human assessment on the fidelity of the output is often considered the "gold-standard" assessment (Ledig et al., 2017; Zhang et al., 2018b). Therefore, many techniques (Johnson et al., 2016; Ledig et al., 2017; Isola et al., 2017) have been proposed for making the output images look perceptually pleasing to human.

However, image processing outputs may not be accurately recognized by image recognition systems. As shown in Fig. 1, the output image of an denoising model could easily be recognized by a human as a bird, but a recognition model classifies it as a kite. One could specifically train a recognition model only on these output images produced by the denoising model to achieve better performance on such images, or could leverage some domain adaptation approaches to adapt the

recognition model to this domain, but the performance on natural images can be harmed. This retraining/adaptation scheme might also be impractical considering the significant overhead induced by catering to various image processing tasks and models.

With the fast-growing size of image data, many images are often "viewed" and analyzed more by machines than by humans. Nowadays, any image uploaded to the Internet is likely to be analyzed by certain vision systems. For example, Facebook uses a system called Rosetta to extract texts from over 1 billion user-uploaded images every day (Maria, 2018). It is of great importance that the processed images be recognizable by not only humans, but also by machines. In other words, recognition systems (e.g., image classifier or object detector), should be able to accurately explain the underlying semantic meaning of the image content. In this way, we make them potentially easier to search, recommended to more interested audience, and so on, as these procedures are mostly executed by machines based on their understanding of the images. Therefore, we argue that image processing systems should also aim at better machine recognizability. We call this problem "Recognition-Aware Image Processing".

It is also important that the enhanced recognizability is not specific to any concrete neural network-based recognition model, i.e., the improvement on recognition performance is only achieved when the output images are evaluated on that particular model. Instead, the improvement should ideally be transferable when evaluated on different models, to support its usage without access to possible future recognition systems, since we may not decide what model will be used for recognizing the processed image, for example if we upload it to the Internet or share it on social media. We may not know what network architectures (e.g. ResNet or VGG) will be used for inference, what object categories the downstream model recognizes (e.g. animals or scenes), or even what task will be performed on the processed image (e.g. classification or detection). Without these specifications, it might be hard to enhance image's machine semantics.

In this work, we propose simple and highly effective approaches to make image processing outputs more accurately recognized by downstream recognition systems, transferable among different recognition architectures, categories and tasks. The approaches we investigate add a recognition loss optimized jointly with the image processing loss. The recognition loss is computed using a fixed recognition model that is pretrained on natural images, and can be done in an unsupervised manner, e.g., without semantic labels of the image. It can be optimized either directly by the original image processing network, or through an intermediate transforming network. We conduct extensive experiments, on multiple image enhancement/restoration (super-resolution, denoising, and JPEG-deblocking) and recognition (classification and detection) tasks, and demonstrate that our approaches can substantially boost the recognition accuracy on the downstream systems, with minimal or no loss in the image processing quality measured by conventional metrics. Also, the accuracy improvement transfers favorably among different recognition model architectures, object categories, and recognition tasks, which renders our simple solution effective even when we do not have access to the downstream recognition models. Our contributions can be summarized as follows:

- We propose to study the problem of enhancing the machine interpretability of image processing outputs, a desired property considering the amount of images analyzed by machines nowadays.

- We propose simple and effective methods towards this goal, suitable for different use cases, e.g., without ground truth semantic labels. Extensive experiments are conducted on multiple image processing and recognition tasks, demonstrating the wide applicability of the proposed methods.

- We show that using our simple approaches, the recognition accuracy improvement could transfer among recognition architectures, categories and tasks, a desirable behavior making the proposed methods applicable without access to the downstream recognition model.

## 2 RELATED WORK

Image processing/enhancement problems such as super-resolution and denoising have a long history (Tsai, 1984; Park et al., 2003; Rudin et al., 1992; Candès et al., 2006). Since the initial success of deep neural networks on these problems (Dong et al., 2014; Xie et al., 2012; Wang et al., 2016b), a large body of works try to investigate better model architecture design and training techniques (Dong et al., 2016; Kim et al., 2016b; Shi et al., 2016; Kim et al., 2016a; Mao et al., 2016; Lai et al., 2017; Tai et al., 2017a; Tong et al., 2017; Tai et al., 2017b; Lim et al., 2017; Zhang et al., 2018d; Ahn

et al., 2018; Lefkimmiatis, 2018; Chen et al., 2018; Haris et al., 2018b), mostly on the image super-resolution task. These works focus on generating high visual quality images under conventional metrics or human evaluation, without considering recognition performance on the output.

There are also a number of works that relate image recognition with processing. Some works (Zhang et al., 2016; Larsson et al., 2016; Zhang et al., 2018c; Sajjadi et al., 2017) use image classification accuracy as an evaluation metric for image colorization/super-resolution, but without optimizing for it during training. Wang et al. (2016a) incorporates super-resolution and domain adaptation techniques for better recognition on very low resolution images. Bai et al. (2018) train a super-resolution and refinement network simultaneously to better detect faces in the wild. Zhang et al. (2018a) train networks for face hallucination and recognition jointly to achieve better recover the face identity from low-resolution images. Liu et al. (2018) considers 3D face reconstruction and trains the recognition model jointly with the reconstructor. Sharma et al. (2018) trains a classification model together with an enhancement module. Our problem setting is different from these works, in that we assume we do not have the control on the recognition model, as it might be on the cloud or decided in the future, thus we advocate adapting the image processing model only. This also ensures the recognition model is not harmed on natural images. Haris et al. (2018a) investigate how super-resolution could help object detection in low-resolution images. VidalMata et al. (2019) and Banerjee et al. (2019) also aims to enhance machine accuracy on poor-conditioned images but mostly focus on better image processing techniques without using recognition models. Wang et al. (2019) propose a method to make denoised images more accurately segmented, also presenting some interesting findings in transferability. Most existing works only consider one image processing task or image domain, and develop specific techniques, while our simpler approach is task-agnostic and potentially more widely applicable. Our work is also related but different from those which aims for robustness of the recognition model (Hendrycks & Dietterich, 2019; Li et al., 2019; Shankar et al., 2018), since we focus on the training of the processing models and assume the recognition model is given.

## 3 METHOD

In this section we first introduce the problem setting of "recognition-aware" image processing, and then we develop various approaches to address it, each suited for different use cases.

### 3.1 PROBLEM SETTING

In a typical image processing problem, given a set of training input images $\{I_{in}^k\}$ and corresponding target images $\{I_{target}^k\}$ ($k = 1, \cdots N$), we aim to train a neural network that maps an input image to its corresponding target. For example, in image denoising, $I_{in}^k$ is a noisy image and $I_{target}^k$ is the corresponding clean image. Denoting this mapping network as $P$ (for processing), parameterized by $W_P$, during training our optimization objective is:

$$\min_{W_P} L_{proc} = \frac{1}{N} \sum_{k=1}^{N} l_{proc} \left( P\left(I_{in}^k\right), I_{target}^k \right), \tag{1}$$

where $P\left(I_{in}^k\right)$ is simply the output of the processing model $I_{out}$, and $l_{proc}$ is the loss function for each sample. The pixel-wise mean-squared-error (MSE, or $L_2$) loss is one of the most popular choices. During evaluation, the performance is typically measured by average similarity (e.g., PSNR, SSIM) between $I_{target}^k$ and $I_{out}^k = P\left(I_{in}^k\right)$, or through human assessment.

In our problem setting of recognition-aware processing, we are interested in a recognition task, with a trained recognition model $R$ ($R$ for recognition), parameterized by $W_R$. We assume each input/target image pair $I_{in}^k/I_{target}^k$ is associated with a ground truth semantic label $S^k$ for the recognition task. Our goal is to train a image processing model $P$ such that the recognition performance on the output images $\{I_{out}^k = P\left(I_{in}^k\right)\}$ is high, when evaluated using $R$ with the semantic labels $\{S^k\}$. In practice, the recognition model $R$ might not be available (e.g., on the cloud), in which case we could resort to other models if the performance improvement transfers among models.

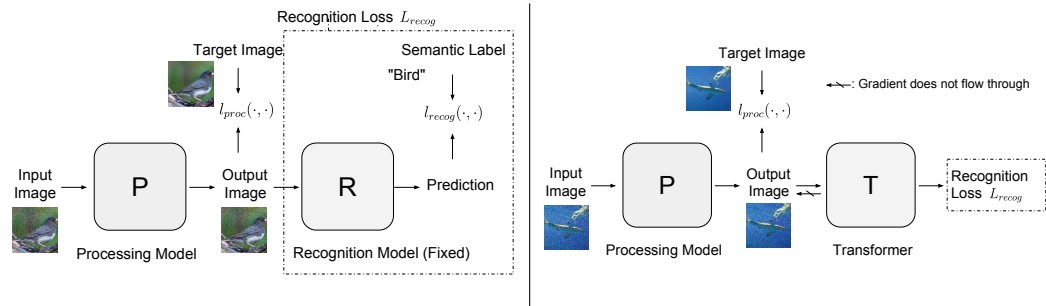

Figure 2: *Left*: RA (Recognition-Aware) processing. In addition to the image processing loss, we add a recognition loss using a fixed recognition model $R$, for the processing model $P$ to optimize. *Right*: RA with transformer. "Recognition Loss" stands for the dashed box in the left figure. A Transformer $T$ is introduced between the output of $P$ and input of $R$, to optimize recognition loss. We cut the gradient from recognition loss flowing to $P$, such that $P$ only optimizes the image processing loss and the image quality is guaranteed not affected.

## 3.2 Optimizing Recognition Loss

Given our goal is to make the output images by $P$ more recognizable by $R$, it is natural to add a recognition loss on top of the objective of the image processing task (Eqn. 1) during training:

$$\min_{W_P} L_{recog} = \frac{1}{N} \sum_{k=1}^{N} l_{recog} \left( R \left( P \left( I_{in}^k \right) \right), S^k \right) \tag{2}$$

$l_{recog}$ is the per-example recognition loss defined by the downstream recognition task. For example, for image classification, $l_{recog}$ could be the cross-entropy (CE) loss. Adding the image processing loss (Eqn. 1) and recognition loss (Eqn. 2) together, our total training objective becomes

$$\min_{W_P} L_{proc} + \lambda L_{recog} \tag{3}$$

where $\lambda$ is the coefficient controlling the weights of $L_{recog}$ relative to $L_{proc}$. We denote this simple solution as "RA (Recognition-Aware) processing", which is visualized in Fig. 2 left. Note that once the training is finished, the recognition model used as loss is not needed anymore, and during inference, we only need the processing model P, thus no additional overhead is introduced when the model is actually put to deployment.

A potential shortcoming of directly optimizing $L_{recog}$ is that it might deviate $P$ from optimizing the original loss $L_{proc}$, and the trained $P$ will generate images that are not as good as if we only optimize $L_{proc}$. We will show that in experiments, however, with proper choice of $\lambda$, we could substantially boost the recognition performance with minimal or no sacrifice on image quality.

If using $R$ as a fixed loss function can only boost the recognition accuracy on $R$ itself, the use of the method could be restricted. Sometimes we do not have the knowledge about the downstream recognition model or even task, but we still would like to improve future recognition performance. Interestingly, we find that image processing models trained with the loss of one recognition model $R_1$, can also boost the performance when evaluated using recognition model $R_2$, even if model $R_2$ has a different architecture, recognizes a different set of categories or even is trained for a different task. This makes our method effective even when we cannot access the target downstream model, in which case we could use another trained model we do have access to as the loss function. This phenomenon also implies that the "recognizability" of a processed image can be a more general notion than just the extent it fits to a specific model. More details on how the improvement is transferable among different recognition models will be presented in the experiments.

## 3.3 Unsupervised Optimization of Recognition Loss

The solution mentioned above requires semantic labels available for training images, which however, may not be satisfied all the time. In this case, we could instead resort to regress the recognition model's output of the target image $R(I_{target}^k)$, given the target images $\{I_{target}^k\}$ at hand, and that the recognition model $R$ is pretrained and fixed. The recognition objective in Eqn. 2 changes to

$$\min_{W_P} L_{recog} = \frac{1}{N} \sum_{k=1}^{N} l_{dis} \left( R \left( P \left( I_{in}^k \right) \right), R \left( I_{target}^k \right) \right) \tag{4}$$

where $l_{dis}$ is a distance metric between $R$'s output given input of processed image $P\left(I_{in}^k\right)$ and ground truth target image $I_{target}^k$. For example, when $R$ is a classification model and outputs a probability distribution over classes, $l_{dis}$ could be the KL divergence or simply a $L_2$ distance. During evaluation, the output of $R$ is still compared to the ground truth semantic label $S^k$. We call this approach "unsupervised RA". Note that it is only "unsupervised" for training model $P$, but not necessarily for the model $R$. The (pre)training of the model $R$ is not our concern since in our problem setting (Sec. 3.1) $R$ is a given trained model, and it can be trained in any manner, either with or without full supervision, and it can even be trained with another dataset as we later show in Sec. 4.4. This approach is to some extent related to the "knowledge distillation" paradigm (Hinton et al., 2014) used for network model compression, where the output of a large model is used to guide the output of a small model, given the same input images. Instead we use the same recognition model $R$ but guide the upstream processing model to generate input to $R$ which produces similar output with that of the target image.

### 3.4 Using an Intermediate Transformer

Sometimes we do want to guarantee that the added recognition loss $L_{recog}$ will not deviate the model $P$ from optimizing its original loss. We can achieve this by introducing another intermediate transformation model $T$. After the input image going through the image processing model $P$, the output image is first fed to the model $T$, and $T$'s output image serves as the input for the recognition model $R$ (Fig. 2 right). In this case, $T$'s parameters $W_T$ are optimized for minimizing the recognition loss:

$$\min_{W_T} L_{recog} = \frac{1}{N} \sum_{k=1}^{N} l_{recog}\left(R\left(T\left(P\left(I_{in}^k\right)\right)\right), S^k\right) \tag{5}$$

In this way, with the help of $T$ on optimizing the recognition loss, the model $P$ can now "focus on" its original image processing loss $L_{proc}$. The optimization objective becomes:

$$\min_{W_P} L_{proc} + \min_{W_T} \lambda L_{recog} \tag{6}$$

In Eqn. 6, $P$ is still solely optimizing $L_{proc}$ as in the original image processing problem (Eqn. 1). $P$ is learned as if there is no recognition loss, and therefore the image processing quality of its output will not be affected. This could be achieved by "cutting" the gradient generated by $L_{recog}$ between the model $T$ and $P$ (Fig. 2 right). The responsibility for a better recognition performance falls on the model $T$. We term this solution as "RA with transformer".

The downside of using a transformer compared with directly optimizing recognition loss using the processing model, is that there are two instances for each image (the output of model $P$ and $T$), one is "for human" and the other is "for machines". Also, as we will show later, it can sometimes harm the transferability of the performance improvement, possibly because there is no image processing loss as a constraint on $T$'s output. Therefore, the transformer is best suited for the case where we want to guarantee the image processing quality not affected at all, at the expense of maintaining another image and losing some transferability.

## 4 Experiments

We evaluate our proposed methods on three image processing tasks, namely image super-resolution, denoising, and JPEG-deblocking. More specifically, these are image enhancement or restoration tasks, where usually the target image is an enhanced image or the original image. Other more broader image processing tasks such as pattern detection, segmentation, object extraction are not considered in this work. To obtain the input images, for super-resolution, we use a downsampling scale factor of $4\times$; for denoising, we add Gaussian noise on the images with a standard deviation of 0.1 to obtain the noisy images; for JPEG deblocking, a quality factor of 10 is used to compress the image to JPEG format. We pair these three tasks with two common visual recognition tasks, image classification and object detection. We adopt the SRResNet (Ledig et al., 2017) as the architecture of the image processing model $P$, due to its popularity and simplicity. For the transformer model $T$, we use the 6-block ResNet architecture in CycleGAN (Zhu et al., 2017), a general-purpose image to image transformation network. For classification we use the ImageNet and for detection we use PASCAL VOC as our benchmark. The recognition architectures are ResNet, VGG and DenseNet. Training is performed with the training set and results on the validation set are reported. For more details on the training settings and hyperparameters of each task, please refer to Appendix A.

## 4.1 Evaluation on the Same Recognition Model

| Task | Super-resolution | | | | | Denoising | | | | | JPEG-deblocking | | | | |
|---|---|---|---|---|---|---|---|---|---|---|---|---|---|---|---|
| Classification Model | R18 | R50 | R101 | D121 | V16 | R18 | R50 | R101 | D121 | V16 | R18 | R50 | R101 | D121 | V16 |
| No Processing | 46.3 | 50.4 | 55.5 | 51.6 | 42.1 | 46.8 | 55.8 | 61.3 | 59.7 | 46.7 | 43.1 | 47.7 | 55.2 | 49.2 | 43.9 |
| Plain Processing | 52.6 | 58.8 | 61.9 | 57.7 | 50.2 | 61.9 | 68.0 | 69.1 | 66.4 | 60.9 | 48.2 | 53.8 | 56.0 | 52.9 | 42.4 |
| RA Processing | 61.8 | 67.3 | 69.6 | 66.0 | 61.9 | 65.1 | **71.2** | **72.7** | **69.8** | **66.5** | 57.7 | 63.6 | 65.8 | 62.3 | 56.7 |
| Unsupervised RA | 61.3 | 66.9 | 69.4 | 65.3 | 61.0 | 61.7 | 68.6 | 70.8 | 67.1 | 63.6 | 53.8 | 60.4 | 63.4 | 59.7 | 53.1 |
| RA w/ Transformer | **63.0** | **68.2** | **70.1** | **66.5** | **63.0** | **65.2** | 70.9 | 72.3 | 69.6 | 65.9 | **59.8** | **65.1** | **66.7** | **63.9** | **58.7** |

(a) Accuracy (%) on ImageNet classification. The five models achieve 69.8, 76.2, 77.4, 74.7, 73.4 on original images.

| Task | Super-resolution | | | | Denoising | | | | JPEG-deblocking | | | |
|---|---|---|---|---|---|---|---|---|---|---|---|---|
| Detection Model | R18 | R50 | R101 | V16 | R18 | R50 | R101 | V16 | R18 | R50 | R101 | V16 |
| No Processing | 67.9 | 70.3 | 72.1 | 63.6 | 51.8 | 56.5 | 61.8 | 38.9 | 49.3 | 54.5 | 64.1 | 38.4 |
| Plain Processing | 69.2 | 70.7 | 73.3 | 64.2 | 68.9 | 72.0 | 74.7 | 65.8 | 63.7 | 66.5 | 70.4 | 60.3 |
| RA Processing | 71.2 | **74.4** | **75.6** | **68.1** | 70.9 | 73.7 | 75.6 | 67.6 | 67.4 | 70.4 | 72.9 | 63.9 |
| RA w/ Transformer | **71.4** | 74.2 | **75.6** | 66.0 | **71.0** | **73.9** | **75.9** | **67.7** | **68.5** | **70.7** | **73.7** | **64.4** |

(b) mAP on PASCAL VOC object detection. The four models achieve 74.2, 76.8, 77.9, 72.2 on original images.

Table 1: Recognition-Aware (RA) processing techniques can substantially boost the recognition accuracy.

We first show our results when evaluating on the same recognition model, i.e., the $R$ used for evaluation is the same as the $R$ we use as the recognition loss in training. Table 1a shows our results on ImageNet classification. ImageNet-pretrained classification models ResNet-18/50/101, DenseNet-121 and VGG-16 are denoted as R18/50/101, D121, V16 in Table 1a. The "No Processing" row denotes the recognition performance on the input of the image processing model: for denoising/JPEG-deblocking, this corresponds to the noisy/JPEG-compressed images; for super-resolution, the low-resolution images are bicubic interpolated to the original resolution. "Plain Processing" denotes conventional image processing models trained without recognition loss as described in Eqn. 1. We observe that a plainly trained processing model can boost the accuracy over unprocessed images. These two are considered as baselines in our experiments.

From Table 1a, using RA processing can significantly boost the accuracy of output images over plainly processed ones, for all image processing tasks and recognition models. This is more prominent when the accuracy of plain processing is lower, e.g., in super-resolution and JPEG-deblocking, in which case we mostly obtain ~10% accuracy improvement. Even without semantic labels, our unsupervised RA can still in most cases outperform baseline methods, despite achieves lower accuracy than its supervised counterpart. Also in super-resolution and JPEG-deblocking, using an intermediate transformer $T$ can bring additional improvement over RA processing.

The results for object detection are shown in Table 1b. We observe similar trend as in classification: using recognition loss can consistently improve the mAP over plain image processing by a notable margin. On super-resolution, RA processing mostly performs on par with RA with transformer, but on the other two tasks using a transformer is slightly better. The model with transformer performs better more often possibly because with this extra network in the middle, the capacity of the whole system is increased: in RA Processing the processing model P optimizes both processing and recognition loss, but now P optimizes processing loss while T optimizes recognition loss

## 4.2 Transfer between Recognition Architectures

In reality, sometimes the recognition model $R$ we want to eventually evaluate the output images on might not be available for us to use as a loss for training, e.g., it could be on the cloud, kept confidential or decided later. In this case, we could train an image processing model $P$ using recognition model $R_A$ that is accessible to us, and after we obtain the trained model $P$, evaluate its output images' recognition accuracy using another unseen recognition model $R_B$. We evaluate all model architecture pairs on ImageNet classification in Table 2 and Table 3, for RA Processing and RA with Transformer respectively, where row corresponds to the model used as recognition loss ($R_A$), and

column corresponds to the evaluation model ($R_B$). For RA with Transformer, we use the processing model $P$ and transformer $T$ trained with $R_A$ together when evaluating on $R_B$.

| Task | Super-resolution | | | | | Denoising | | | | | JPEG-deblocking | | | | |
|---|---|---|---|---|---|---|---|---|---|---|---|---|---|---|---|
| Evaluation on | R18 | R50 | R101 | D121 | V16 | R18 | R50 | R101 | D121 | V16 | R18 | R50 | R101 | D121 | V16 |
| Plain Processing | 52.6 | 58.8 | 61.9 | 57.7 | 50.2 | 61.9 | 68.0 | 69.1 | 66.4 | 60.9 | 48.2 | 53.8 | 56.0 | 52.9 | 42.4 |
| RA w/ R18 | **61.8** | 66.7 | 68.8 | 64.7 | 58.2 | **65.1** | 70.6 | 71.9 | 69.1 | 63.8 | **57.7** | 62.3 | 64.3 | 60.7 | 52.8 |
| RA w/ R50 | 59.3 | **67.3** | 68.8 | 64.3 | 59.1 | 64.2 | **71.2** | 72.2 | 69.2 | 64.7 | 55.8 | **63.6** | 64.7 | 61.0 | 53.5 |
| RA w/ R101 | 58.8 | 66.0 | **69.6** | 63.4 | 58.2 | 64.0 | 70.5 | **72.7** | 68.9 | 64.8 | 54.9 | 61.5 | **65.8** | 60.3 | 52.8 |
| RA w/ D121 | 59.0 | 65.6 | 67.8 | **66.0** | 57.4 | 64.2 | 70.6 | 72.0 | **69.8** | 64.3 | 54.8 | 61.8 | 64.4 | **62.3** | 52.9 |
| RA w/ V16 | 57.9 | 64.8 | 67.0 | 63.0 | **61.9** | 63.9 | 70.4 | 72.0 | 68.8 | **66.5** | 54.5 | 60.9 | 63.1 | 59.7 | **56.7** |

Table 2: Transfer between recognition architectures using RA processing, on ImageNet classification. A image processing model trained with model $R_A$ (row) as recognition loss can improve the recognition performance on model $R_B$ (column) over plain processing.

In Table 2's each column, training with any model $R_A$ produces substantially higher accuracy than plainly processed images on $R_B$. Thus, we conclude that the improvement on the recognition accuracy is transferable among different recognition architectures. A possible explanation for this is that these models are all trained on the same ImageNet dataset, such that their mapping functions from input to output are similar, and optimizing the loss of one would lead to the lower loss of another. This phenomenon enables us to use RA processing without the knowledge of the downstream recognition model architecture. However, among all rows, the $R_A$ that achieves the highest accuracy is still the same model as $R_B$, indicated by the diagonal boldface numbers in Table 2.

| Task | Super-resolution | | | | | Denoising | | | | | JPEG-deblocking | | | | |
|---|---|---|---|---|---|---|---|---|---|---|---|---|---|---|---|
| Evaluation on | R18 | R50 | R101 | D121 | V16 | R18 | R50 | R101 | D121 | V16 | R18 | R50 | R101 | D121 | V16 |
| Plain Processing | 52.6 | 58.8 | 61.9 | 57.7 | 50.2 | 61.9 | 68.0 | 69.1 | 66.4 | 60.9 | 48.2 | 53.8 | 56.0 | 52.9 | 42.4 |
| RA w/$T$ w/ R18 | **63.0** | 59.2 | 67.0 | 63.9 | 27.0 | **65.2** | 69.4 | 71.6 | 68.4 | 40.3 | **59.8** | 58.7 | 62.6 | 60.3 | 19.9 |
| RA w/$T$ w/ R50 | 60.5 | **68.2** | 68.9 | 65.8 | 40.4 | 63.1 | **70.9** | 71.5 | 68.6 | 48.7 | 55.0 | **65.1** | 63.9 | 61.9 | 31.5 |
| RA w/$T$ w/ R101 | 59.6 | 66.2 | **70.1** | 65.1 | 35.6 | 62.4 | 68.8 | **72.3** | 67.6 | 52.3 | 54.8 | 61.3 | **66.7** | 24.8 | 60.5 |
| RA w/$T$ w/ D121 | 58.5 | 64.2 | 66.9 | **66.5** | 27.3 | 58.0 | 66.8 | 67.3 | **69.6** | 46.7 | 46.6 | 57.2 | 59.0 | **63.9** | 9.0 |
| RA w/$T$ w/ V16 | 59.2 | 64.7 | 67.8 | 65.0 | **63.0** | 57.6 | 64.0 | 67.1 | 55.7 | **63.1** | 56.1 | 61.2 | 63.4 | 58.7 | **60.1** |

Table 3: Transfer between architectures using RA with Transformer ($T$), on ImageNet classification.

Meanwhile in Table 3, in most cases improvement is still transferable when we use a transformer $T$, but there are a few exceptions. For example, when $R_A$ is ResNet or DenseNet and when $R_B$ is VGG-16, in most cases the accuracy fall behind plain processing by a large margin. This weaker transferability is possibly caused by the fact that there is no constraint imposed by the image processing loss on $T$'s output, thus it "overfits" more to the specific $R$ it is trained with. For more results on object detection and unsupervised RA, please refer to the Appendix B.1. This is intuitive since the processing model optimizes the same recognition loss during training as that used in evaluation.

One of the reasons why our method attains transferability is possibly that these models learn many common features that could be useful for general computer vision, especially in shallower layers. More importantly, the reason could be similar to the reason why adversarial examples can transfer among models: different models' decision boundaries are similar. Liu et al. (2016) studies adversarial examples' transferability and shows decision boundaries of different models align well with each other; Tramèr et al. (2017) quantitatively analyzes similarity of different models' decision boundaries, and shows that the boundaries are close in arbitrary directions, whether adversarial or benign.

## 4.3 TRANSFER BETWEEN OBJECT CATEGORIES

What if the $R_A$ and $R_B$ recognize different categories of objects? Can RA processing still bring transferable improvement? To answer this question, we divide the 1000 classes from ImageNet into two splits (denoted as category $A/B$), each with 500 classes, and train two 500-way classification models (ResNet-18) on both splits, obtaining $R_A$ and $R_B$. Next, we train two image processing models $P_A$ and $P_B$ with the $R_A$ and $R_B$ as recognition loss, using images from category $A$ and $B$ respectively. Note that neither the image processing model $P$ nor the recognition model $R$ has

seen any images from the other split of categories during training, and $R_A$ and $R_B$ learn completely different mappings from input to output. The plain processing counterparts of $P_A$ and $P_B$ are also trained with category $A$ and $B$ respectively, but without the recognition loss. We evaluate obtained image processing models on both splits, and the results are shown in Table 4.

| Task | Super-resolution | | Denoising | | JPEG-deblocking | |
|---|---|---|---|---|---|---|
| Train/Eval Category | Cat $A$ | Cat $B$ | Cat $A$ | Cat$B$ | Cat $A$ | Cat $B$ |
| Cat $A$ Plain | 59.6 | 60.1 | 67.6 | 68.0 | 54.2 | 55.5 |
| Cat $A$ RA | **67.2** | **66.5** | **69.7** | **69.4** | **63.0** | **62.3** |
| Cat $B$ Plain | 59.6 | 60.2 | 67.0 | 67.5 | 54.7 | 56.0 |
| Cat $B$ RA | **66.4** | **67.8** | **69.4** | **69.7** | **62.1** | **63.5** |

Table 4: Transfer between different object categories (500-way accuracy %). RA processing on one set of categories can also improve the performance on another. "Cat" means category.

We observe that using RA processing still benefits the recognition accuracy even when transferring across categories (e.g., in super-resolution, from 60.1% to 66.5% when transferring from category $A$ to category $B$, on super resolution). The improvement is only marginally lower than directly training with recognition model of the same category (e.g., from 60.2% to 67.8% when trained and evaluated both on category $B$). Such transferability between categories suggest the learned image processing models do not improve accuracy by adding category-specific signals to the output images, instead they generate more general signals that enable a wider set of classes to be better recognized.

## 4.4 TRANSFER BETWEEN RECOGNITION TASKS

What if we take a further step to the case when $R_A$ and $R_B$ not only recognize different categories, but also perform different tasks? We evaluate such task transferability for when task $A$ is classification and task $B$ is object detection in Table 5. For results on the opposite direction and results for unsupervised RA, please refer to Appendix B.2.

| Task | Super-resolution | | | | Denoising | | | | JPEG-deblocking | | | |
|---|---|---|---|---|---|---|---|---|---|---|---|---|
| Evaluation on | R18 | R50 | R101 | V16 | R18 | R50 | R101 | V16 | R18 | R50 | R101 | V16 |
| Plain Processing | 68.5 | 69.7 | 73.1 | 63.2 | 68.1 | 71.6 | 74.1 | 65.7 | 62.4 | 65.6 | 69.5 | 58.3 |
| RA w/ R18 | **71.3** | 73.5 | **75.6** | **67.8** | **70.6** | 73.1 | 75.5 | 64.1 | 67.7 | 70.3 | **73.2** | 62.4 |
| RA w/ R50 | 70.8 | 73.2 | 74.8 | **67.8** | 70.4 | 73.1 | **75.8** | 66.2 | 67.8 | 70.2 | 73.1 | 62.8 |
| RA w/ R101 | 70.7 | 73.2 | 75.3 | 67.0 | 70.5 | **73.5** | 75.7 | 66.9 | **68.1** | 70.2 | 72.8 | 63.2 |
| RA w/ D121 | 71.2 | **73.6** | 75.3 | 67.2 | 70.5 | 73.2 | 75.7 | 65.7 | **68.1** | **70.5** | 73.1 | 62.6 |
| RA w/ V16 | 70.4 | 72.4 | 74.6 | 67.5 | **70.6** | 73.0 | 75.7 | **67.7** | 67.8 | 70.3 | **73.2** | **63.7** |

Table 5: Transfer from ImageNet classification to PASCAL VOC object detection (mAP). Processing model $P$ trained with classification model $A$ (row) can improve the performance on detection model $B$ (column).

In Table 5, note that rows indicate classification models used as loss and columns indicate detection models, so even if they are of the same name (e.g., "R18"), they are still different models, and are trained on different datasets for different tasks. We are transferring between architectures, categories as well as tasks in this experiment. There is even a domain shift since the model $P$ is trained with ImageNet training set but fed with PASCAL VOC input images during evaluation. Here "Plain Processing" models are trained on the ImageNet instead of PASCAL VOC dataset, thus the results are different from those in Table 1b. We observe that except two cases on the "V16" column in denoising, using classification loss on model $A$ (row) can boost the detection accuracy on model $B$ notably upon plain processing. This improvement is even comparable with directly training using the detection loss, as in Table 1b. Such task transferability suggests the "machine semantics" of the image could even be a task-agnostic property, and makes our method even more broadly applicable.

## 4.5 IMAGE PROCESSING QUALITY COMPARISON

We have analyzed the recognition accuracy of the output images, now we compare the output image quality using conventional metrics PSNR and SSIM. When using RA with transformer, the output quality of $P$ is guaranteed unaffected, therefore here we evaluate RA processing. We use ResNet-18 on ImageNet as $R$, and report results with different $\lambda$s (Eqn. 3) in Table 6.

$\lambda = 0$ corresponds to plain processing. When $\lambda = 10^{-4}$, in super-resolution, the PSNR/SSIM metrics are even slightly higher, and in denoising and JPEG-deblocking they are only marginally worse. However, the accuracy obtained is significantly higher. This suggests that the added recognition loss is not harmful when $\lambda$ is chosen properly. When $\lambda$ is excessively

| $\lambda$ | Super-resolution | Denoising | JPE-deblocking |
|---|---|---|---|
| 0 | 26.29/0.795/52.6 | **31.24/0.895**/61.9 | **27.50/0.825**/48.2 |
| $10^{-4}$ | **26.33/0.803**/59.2 | 31.18/0.894/64.4 | **27.50**/0.823/56.0 |
| $10^{-3}$ | 26.31/0.792/**61.8** | 30.78/0.884/**65.1** | 27.17/0.810/**57.7** |
| $10^{-2}$ | 25.47/0.760/61.3 | 29.71/0.855/64.3 | 26.32/0.776/56.6 |

Table 6: PSNR/SSIM/Accuracy when using different $\lambda$s, on ImageNet dataset.

large ($10^{-2}$), the image quality is hurt more, and interestingly even the recognition accuracy start to decrease. A proper balance between image processing loss and recognition loss is needed for both image quality and performance on downstream recognition tasks.

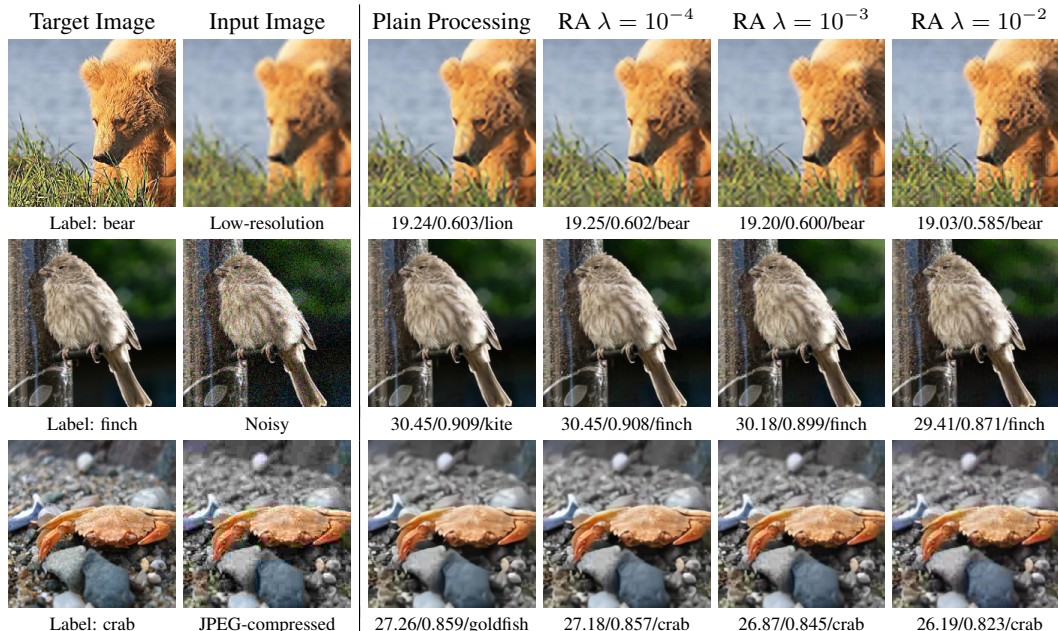

Figure 3: Examples where outputs of RA processing models can be correctly classified but those from plain processing models cannot. PSNR/SSIM/class prediction is shown below each output image. Slight differences between images from plain processing and RA processing models could be noticed when zoomed in.

In Fig. 3, we visualize some examples where the output image is incorrectly classified with a plain image processing model, and correctly recognized with RA processing. With smaller $\lambda$ ($10^{-2}$ and $10^{-3}$), the image is nearly the same as the plainly processed images. When $\lambda$ is too large ($10^{-2}$), we could see some extra textures when zooming in. For more results please refer to Appendix C.

## 5  ANALYSIS

In this section we analyze some alternatives to our approaches. All experiments in this section are conducted using RA processing on super-resolution, with ResNet-18 trained on ImageNet as the recognition model, and $\lambda = 10^{-3}$ if used.

**Training without the Image Processing Loss.** It is possible to train the processing model on the recognition loss $L_{recog}$, without even keeping the original image processing loss $L_{proc}$ (Eqn. 3). This may presumably lead to better recognition performance since the model $P$ can now "focus on" optimizing the recognition loss. However, we found removing the original image processing loss hurts the recognition performance: the accuracy drops from 61.8% to 60.9%; even worse, the SSIM/PSNR metrics drop from 26.33/0.792 to 16.92/0.263, which is reasonable since the image processing loss is not optimized during training. This suggests the original image processing loss is helpful for the recognition accuracy, since it helps the corrupted image to restore to its original form.

**Fine-tuning the Recognition Model.** Instead of fixing the recognition model $R$, we could fine-tune it together with the training of image processing model $P$, to optimize the recognition loss.

Many prior works (Sharma et al., 2018; Bai et al., 2018; Zhang et al., 2018a) do train/fine-tune the recognition model jointly with the image processing model. We use SGD with momentum as $R$'s optimizer, and the final accuracy reaches 63.0%. However, since we do not fix $R$, it becomes a model that specifically recognizes super-resolved images, and we found its performance on original target images drops from 69.8% to 60.5%. Moreover, when transferring the trained $P$ on ResNet-56, the accuracy is 62.4 %, worse than 66.7% when we train with a fixed ResNet-18. We lose some transferability if we do not fix the recognition model $R$.

**Training Recognition Models from Scratch.** Other than fine-tuning a pretrained recognition model $R$, we could first train a super-resolution model, and then train $R$ from scratch on the output images. We achieve 66.1% accuracy on the output images in the validation set, higher than 61.8% in RA processing. However, the accuracy on original clean images drops from 69.8% to 66.1%. Alternatively, we could even train $R$ from scratch on the interpolated low-resolution images, in which case we achieve 66.0% on interpolated validation data but only 50.2% on the original validation data. In summary, training or fine-tuning $R$ to cater the need of super-resolved or interpolated images can harm its performance on the original clean images, and causes additional overhead in storing models. In contrast, using our RA processing technique could boost the accuracy of output images with the performance on original images intact.

## 6 CONCLUSION

We investigated the problem of enhancing the machine interpretability of image processing outputs. We find our simple approach – optimizing with the additional recognition loss during training can significantly boost the recognition accuracy with minimal or no loss in image processing quality. Moreover, such improvement can transfer to recognition architectures, object categories, and vision tasks unseen during training, indicating the enhanced interpretability is not specific to one particular model but generalizable to others. This makes the proposed approach feasible even when the future downstream recognition models are unknown.

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

# APPENDIX

## A  EXPERIMENTAL DETAILS

**General Setup.** We evaluate our proposed methods on three image processing tasks: image super-resolution, denoising, and JPEG-deblocking. In those tasks, the target images are all the original images from the datasets. To obtain the input images, for super-resolution, we use a downsampling scale factor of $4\times$; for denoising, we add Gaussian noise on the images with a standard deviation of 0.1 to obtain the noisy images; for JPEG deblocking, a quality factor of 10 is used to compress the image to JPEG format. The image processing loss used is the mean squared error (MSE, or $L_2$) loss. For the recognition tasks, we consider image classification and object detection, two common tasks in computer vision. In total, we have 6 ($3 \times 2$) task pairs to evaluate.

We adopt the SRResNet (Ledig et al., 2017) as the architecture of the image processing model $P$, which is simple yet effective in optimizing the MSE loss. Even though SRResNet is originally designed for super-resolution, we find it also performs well on denoising and JPEG deblocking when its upscale parameter set to 1 for the same input-output sizes. Throughout the experiments, on both the image processing network and the transformer, we use the Adam optimizer (Kingma & Ba, 2014) with an initial learning rate of $10^{-4}$, following the original SRResNet (Ledig et al., 2017). Our implementation is in PyTorch (Paszke et al., 2017).

**Image Classification.** For image classification, we evaluate our method on the large-scale ImageNet benchmark (Deng et al., 2009). We use five pre-trained image classification models, ResNet-18/50/101 (He et al., 2016), DenseNet-121 (Huang et al., 2017) and VGG-16 (Simonyan & Zisserman, 2015) with BN (Ioffe & Szegedy, 2015) (denoted as R18/50/101, D121, V16 in Table 1a), on which the top-1 accuracy (%) of the original validation images is 69.8, 76.2, 77.4, 74.7, and 73.4 respectively. We train the processing models for 6 epochs on the training set, with a learning rate decay of $10\times$ at epoch 5 and 6, and a batch size of 20. In evaluation, we feed unprocessed validation images to the image processing model, and report the accuracy of the output images evaluated on the pre-trained classification networks. For unsupervised RA, we use $L_2$ distance as the function $l_{dis}$ in Eqn. 4. The hyperparameter $\lambda$ is chosen using super-resolution with the ResNet-18 recognition model, on two small subsets for training/validation from the original large training set. The $\lambda$ chosen for RA processing, RA with transformer, and unsupervised RA is $10^{-3}$, $10^{-2}$ and 10 respectively.

**Object Detection.** For object detection, we evaluate on PASCAL VOC 2007 and 2012 dataset, using Faster-RCNN (Ren et al., 2015) as the recognition model. Our implementation is based on the code from (Yang et al., 2017). Following common practice (Redmon et al., 2016; Ren et al., 2015; Dai et al., 2016), we use VOC 07 and 12 trainval data as the training set, and evaluate on VOC 07 test data. The Faster-RCNN training uses the same hyperparameters in (Yang et al., 2017). For the recognition model's backbone architecture, we evaluate ResNet-18/50/101 and VGG-16 (without BN (Ioffe & Szegedy, 2015)), obtaining mAP of 74.2, 76.8, 77.9, 72.2 on the test set respectively. Given those trained detectors as recognition loss functions, we train the models on the training set for 7 epochs, with a learning rate decay of $10 \times$ at epoch 6 and 7, and a batch size of 1. We report the mean Average Precision (mAP) of processed images in the test set. As in image classification, we use $\lambda = 10^{-3}$ for RA processing, and $\lambda = 10^{-2}$ for RA with transformer.

## B   MORE RESULTS ON TRANSFERABILITY

We present some additional results on transferability here.

### B.1   TRANSFERRING BETWEEN ARCHITECTURES

| Task | Super-resolution | | | | Denoising | | | | JPEG-deblocking | | | |
|---|---|---|---|---|---|---|---|---|---|---|---|---|
| Evaluation on | R18 | R50 | R101 | V16 | R18 | R50 | R101 | V16 | R18 | R50 | R101 | V16 |
| Plain Processing | 69.2 | 70.7 | 73.3 | 64.2 | 68.9 | 72.0 | 74.7 | 65.8 | 63.7 | 66.5 | 70.4 | 60.3 |
| RA w/ R18 | **71.2** | 73.8 | 75.2 | 66.9 | **70.9** | **74.0** | 75.5 | 67.2 | **67.4** | 70.0 | 72.3 | 63.5 |
| RA w/ R50 | 70.6 | **74.4** | 75.4 | 66.4 | 70.6 | 73.7 | 75.5 | 67.2 | 67.0 | **70.4** | 72.4 | 63.2 |
| RA w/ R101 | 71.1 | 73.8 | **75.6** | 65.8 | 70.3 | 73.6 | **75.6** | 66.2 | 65.9 | 69.3 | **72.9** | 61.3 |
| RA w/ V16 | 70.4 | 72.8 | 74.9 | **68.1** | 69.9 | 73.4 | **75.6** | **67.6** | 66.1 | 69.3 | 72.1 | **63.9** |

Table 7: Transfer between recognition architectures, evaluated on PASCAL VOC object detection (mAP).

We provide the model transferability results of RA processing on object detection in Table 7. Rows indicate the models trained as recognition loss and columns indicate the evaluation models. We see similar trend as in classification (Table 1a): using other architectures as loss can also improve recognition performance over plain processing; the loss model that achieves the highest performance is mostly the model itself, as can be seen from the fact that most boldface numbers are on the diagonals.

As a complement in Section 4.2, we present the results when transferring between recognition architectures, using unsupervised RA, in Table 8. We note that for super-resolution and JPEG-deblocking, similar trend holds as in (supervised) RA processing, as using any architecture in training will improve over plain processing. But for denoising, this is not always the case. Some models $P$ trained with unsupervised RA are slightly worse than the plain processing counterpart. A possible reason for this is the noise level in our experiments is not large enough and plain processing achieve very high accuracy already.

| Task | Super-resolution | | | | | Denoising | | | | | JPEG-deblocking | | | | |
|------|------|------|------|------|------|------|------|------|------|------|------|------|------|------|------|
| Evaluation on | R18 | R50 | R101 | D121 | V16 | R18 | R50 | R101 | D121 | V16 | R18 | R50 | R101 | D121 | V16 |
| Plain Processing | 52.6 | 58.8 | 61.9 | 57.7 | 50.2 | **61.9** | 68.0 | 69.1 | 66.4 | 60.9 | 48.2 | 53.8 | 56.0 | 52.9 | 42.4 |
| Unsup. RA w/ R18 | **61.3** | 66.3 | 68.6 | 64.5 | 57.3 | 61.7 | 67.9 | 69.7 | 66.4 | 60.5 | **53.8** | 59.1 | 62.0 | 57.5 | 50.0 |
| Unsup. RA w/ R50 | 58.9 | **66.9** | 68.6 | 64.1 | 58.2 | 61.2 | **68.6** | 70.3 | 66.6 | 61.3 | 52.8 | **60.4** | 62.5 | 58.3 | 50.3 |
| Unsup. RA w/ R101 | 57.8 | 64.9 | **69.0** | 62.9 | 56.9 | 60.6 | 68.0 | **70.7** | 66.3 | 60.7 | 52.3 | 58.7 | **63.4** | 57.9 | 49.0 |
| Unsup. RA w/ D121 | 58.0 | 64.7 | 67.2 | **65.3** | 56.0 | 60.7 | 67.8 | 69.7 | **67.1** | 60.3 | 52.2 | 59.2 | 62.2 | **59.7** | 49.9 |
| Unsup. RA w/ V16 | 57.7 | 64.6 | 67.3 | 63.2 | **61.0** | 60.4 | 67.1 | 69.6 | 65.9 | **63.6** | 52.0 | 58.4 | 61.5 | 57.4 | **53.1** |

Table 8: Transfer between recognition architectures using unsupervised RA, on ImageNet classification.

## B.2 Transferring between Recognition Tasks

In Section 4.4, we investigated the transferability of improvement from classification to detection. Here we evaluate the opposite direction, from detection to classification. The results are shown in Table 9. Here, using RA processing can still consistently improve over plain processing for any pair of models, but we note that the improvement is not as significant as directly training using classification models as loss (Table 1a and Table 2).

| Task | Super-resolution | | | | | Denoising | | | | | JPEG-deblocking | | | | |
|------|------|------|------|------|------|------|------|------|------|------|------|------|------|------|------|
| Evaluation on | R18 | R50 | R101 | D121 | V16 | R18 | R50 | R101 | D121 | V16 | R18 | R50 | R101 | D121 | V16 |
| Plain Processing | 53.0 | 58.9 | 62.0 | 57.3 | 50.9 | 59.7 | 65.1 | 67.3 | 63.9 | 59.2 | 48.8 | 54.6 | 56.8 | 53.1 | 44.7 |
| RA w/ R18 | **54.6** | 60.2 | 63.4 | 58.8 | **52.7** | 60.8 | 66.7 | 68.8 | **65.2** | **61.1** | 50.8 | **57.2** | 59.6 | 55.4 | **48.5** |
| RA w/ R50 | 54.0 | 59.7 | 63.0 | 58.7 | 52.0 | 60.5 | 66.6 | 68.5 | 64.9 | 60.8 | 50.7 | 56.9 | 59.2 | 55.3 | 48.3 |
| RA w/ R101 | 54.1 | 59.8 | 63.3 | 58.7 | 52.5 | 60.2 | 66.1 | 68.3 | 64.6 | 60.6 | **51.3** | **57.2** | 59.5 | **55.5** | 48.3 |
| RA w/ V16 | 54.5 | **60.4** | **63.6** | **59.1** | **52.7** | 60.4 | 66.6 | 68.4 | 64.7 | 60.6 | 50.6 | 56.5 | 58.7 | 54.9 | 47.9 |

Table 9: Transfer from PASCAL VOC object detection to ImageNet classification (accuracy %). A image processing model $P$ trained with detection model $A$ (row) as recognition loss can improve the performance on classification model $B$ (column) over plain processing.

Additionally, the results when we transfer the model $P$ trained with unsupervised RA with image classification to object detection are shown in Table 10. In most cases, it improves over plain processing, but for image denoising, this is not always the case. Similar to results in Table 8, this could be because the noise level is relatively low in our experiments.

| | Super-resolution | | | | Denoising | | | | JPEG-deblocking | | | |
|------|------|------|------|------|------|------|------|------|------|------|------|------|
| Evaluation on | R18 | R50 | R101 | V16 | R18 | R50 | R101 | V16 | R18 | R50 | R101 | V16 |
| Plain Processing | 68.5 | 69.7 | 73.1 | 63.2 | 68.1 | 71.6 | 74.1 | 65.7 | 62.4 | 65.6 | 69.5 | 58.3 |
| Unsup. RA w/ R18 | **71.3** | **73.4** | **75.3** | 66.8 | **69.0** | 71.3 | 74.3 | 61.1 | 65.2 | 68.1 | 71.3 | 59.8 |
| Unsup. RA w/ R50 | 70.7 | 73.3 | 75.0 | 66.6 | 68.9 | **71.7** | **74.4** | 63.1 | 65.4 | 68.5 | 71.2 | 60.0 |
| Unsup. RA w/ R101 | 70.7 | 73.2 | 75.0 | 66.2 | 68.9 | 71.3 | 73.9 | 63.3 | 65.2 | 67.9 | 71.1 | 59.6 |
| Unsup. RA w/ D121 | 71.0 | 73.2 | 75.1 | 66.6 | 68.7 | 70.3 | 73.0 | **63.8** | **65.9** | **68.6** | 71.4 | **61.1** |
| Unsup. RA w/ V16 | 70.3 | 72.3 | 74.3 | **67.0** | 68.5 | 70.7 | 74.0 | 63.6 | **65.9** | 68.2 | **71.5** | **61.1** |

Table 10: Transfer from ImageNet classification to PASCAL VOC object detection, using unsupervised RA.

## C  More Visualizations

We provide more visualizations in Fig. 4 where the output image is incorrectly classified by ResNet-18 with a plain image processing model, and correctly recognized with RA processing, as in Fig. 3 at Section 4.5.

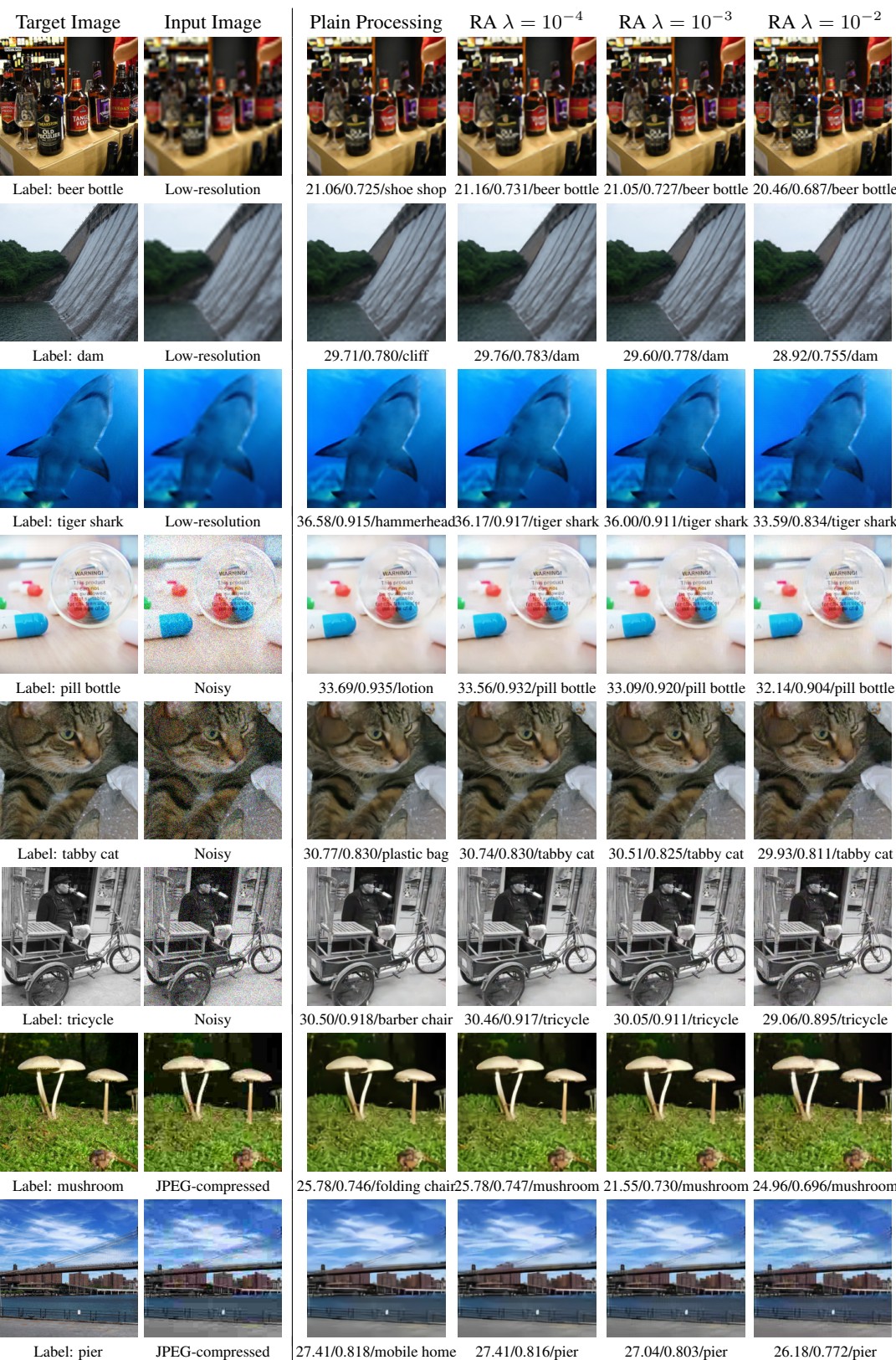

Figure 4: Examples where output images from RA processing models can be correctly classified but those from plain processing models cannot. PSNR/SSIM/class prediction is shown below each output image. Slight differences between images from plain processing and RA processing models (especially with large λs) could be noticed when zoomed in.

# D    RESULTS ON IMAGENET-C

We evaluate our methods on the ImageNet-C benchmark (Hendrycks & Dietterich, 2019). It imposes 17 different types of corruptions on the ImageNet (Deng et al., 2009) validation set. Despite ImageNet-C benchmark is designed for more robust recognition models, but not for testing image processing models, it is a good testbed to test our methods in a broader range of processing tasks. Since only corrupted images from the validation set are released, we divide it evenly for each class into two halves and train/test on its first/second half. The corrupted image is the input image to the processing model and the original clean image is the target image. The recognition model used in this experiment is an ImageNet-pretrained ResNet-18.

In Table 11, we evaluate RA Processing on all 17 types of corruptions, with corruption level 5 as in (Hendrycks & Dietterich, 2019). We observe that RA Processing brings consistent improvement over plain processing, sometimes by an even larger margin than the tasks considered in Sec. 4.

| Type | orig | brit | contr | defoc | elast | gau_b | gau_n | glass | impul | jpeg | motn | pixel | shot | satr | snow | spat | speck | zoom |
|---|---|---|---|---|---|---|---|---|---|---|---|---|---|---|---|---|---|---|
| No Processing | 69.9 | 51.3 | 3.3 | 11.3 | 17.1 | 9.3 | 1.2 | 8.7 | 1.0 | 29.4 | 11.1 | 23.1 | 1.8 | 39.5 | 10.7 | 19.1 | 7.7 | 17.6 |
| Plain Processing | N/A | 59.9 | 18.3 | 25.3 | 18.9 | 21.5 | 21.8 | 20.1 | 24.1 | 43.0 | 42.4 | 50.1 | 24.9 | 54.4 | 34.5 | 60.8 | 36.6 | 17.0 |
| RA Processing | N/A | 61.4 | 30.7 | 33.8 | 35.4 | 27.0 | 32.8 | 25.3 | 35.1 | 46.1 | 48.2 | 54.0 | 35.2 | 57.1 | 43.7 | 63.0 | 45.2 | 31.9 |

Table 11: ImageNet-C results (top-1 accuracy %) under different types of corruptions with corruption level 5.

In Table 12, we experiment with different levels of corruptions with corruption type "speckle noise" and "snow". We also evaluate with our variants – Unsupervised RA and RA with Transformer. We observe that when the corruption level is higher, our methods tend to bring more recognition accuracy gain.

| Corruption Type | Snow | | | | | Speckle noise | | | | |
|---|---|---|---|---|---|---|---|---|---|---|
| Corruption Level | 1 | 2 | 3 | 4 | 5 | 1 | 2 | 3 | 4 | 5 |
| No Processing | 46.7 | 23.6 | 28.0 | 17.6 | 10.7 | 50.5 | 42.8 | 22.9 | 14.5 | 7.7 |
| Plain Processing | 57.1 | 45.1 | 46.0 | 37.1 | 34.5 | 60.3 | 57.0 | 48.4 | 43.2 | 36.6 |
| RA Processing | 60.3 | 51.7 | 51.7 | 45.7 | 43.7 | 62.7 | 60.8 | 54.2 | 50.3 | 45.2 |
| Unsupervised RA | 60.2 | 51.3 | 50.6 | 43.6 | 41.5 | 62.9 | 60.5 | 53.8 | 49.4 | 43.9 |
| RA w/ Transformer | 55.7 | 46.7 | 48.1 | 42.7 | 40.9 | 59.0 | 57.7 | 52.2 | 49.2 | 44.7 |

Table 12: ImageNet-C results (top-1 accuracy %) under different levels of corruptions, with corruption level "snow" and "speckle noise".

In Table 13, we examine the transferability of RA Processing between recognition architectures, using the same two tasks "speckle noise" and "snow", with corruption level 5. Note the recognition loss used during training is from a ResNet-18, and we evaluate the improvement over plain processing on ResNet-50/101, DenseNet-121 and VGG-16. We observe that the improvement over plain processing is transferable among different architectures.

| Corruption Type | Snow | | | | | Speckle noise | | | | |
|---|---|---|---|---|---|---|---|---|---|---|
| Evaluation on | R18 | R50 | R101 | D121 | V16 | R18 | R50 | R101 | D121 | V16 |
| No Processing | 10.7 | 16.6 | 20.9 | 21.7 | 10.5 | 7.7 | 11.7 | 14.5 | 18.6 | 7.1 |
| Plain Processing | 34.5 | 39.1 | 44.6 | 41.1 | 27.4 | 36.6 | 42.4 | 47.7 | 43.0 | 31.3 |
| RA w/ R18 | 43.7 | 47.9 | 51.7 | 47.9 | 37.4 | 45.2 | 50.3 | 53.3 | 49.1 | 39.0 |

Table 13: ImageNet-C results (top-1 accuracy %) under different levels of corruptions, with corruption level "snow" and "speckle noise".

