# OpenReview forum: "Transferable Recognition-Aware Image Processing"
_ICLR.cc/2020/Conference — Reject_

### Official Review · AnonReviewer3 · 2019-10-09
**Official Blind Review #3**

**Rating:** 8

**Review:**

Claims:

The paper presents a concept of "recognition-aware (RA) image processing": when one enhances image in a some way, not only human judjement should be taken into account, but also performance of various computer vision application using that image.

As an example of processing tasks, authors take super-resolution, denoising and JPEG-artifacts removal. Downstream applications covered are image classification and object detection.

Authors propose a several training schemas to solve this problem and discuss a limitations of each one:
 - "simple" preprocessing, when the only image enhancement loss is optimized
 - "RA" joint optimization of recognition and enhancement loss (supervised and unsupervised)
 - a variant when two images are created: one for human and one for machine.

****

Recommendation: strong accept

****
Comments:

 Experiments are vast and performed on a variety of CNN architectures: ResNets, DenseNet ant VGGNet.
 Because one cannot predict, which computer vision tasks will be needed in the future, the natural question arise: how the results got for one set of tasks, architectures and image enhancement types transfer to another. Paper carefully studies this aspect as well.

 Overall paper is well written and is pleasure to read. While reading, I made notes to ask in review - just to see the my questions answered in a next section.
Authors also provide source code for training. I haven`t run them though, but glanced through them.

 Weaknesses: I cannot really find a significant one. As a minor points:
  - I would recommend to cite not the last papers for image enhancement porblems themselves like super-resolution and denoising: these are old problems with rich history, e.g.

 L. Rudin, S. Osher, and E. Fatemi, Nonlinear total variation based noise removal algorithms Physica D, 60 (1992), pp. 259–268.

 - "Transformer" is probably bad name for deep learning component, as it is already widely used for a specific seq2seq architecture


****
After rebuttal: I am now even more convinced that paper should be accepted.

**Experience Assessment:**

I have published one or two papers in this area.

**Review Assessment: Checking Correctness Of Derivations And Theory:**

N/A

**Review Assessment: Checking Correctness Of Experiments:**

I carefully checked the experiments.

**Review Assessment: Thoroughness In Paper Reading:**

I read the paper thoroughly.

---

> ### Author Response · Authors · 2019-11-10
> **Response to AnonReviewer3**
>
> Thank you for your positive feedback! We are glad to see your acknowledgement on our contributions, and we are happy address your concerns below:
>
> 1. In the updated revision, we’ve added a few citations of classic papers on super-resolution and denoising, in the first sentence of related work: “Image processing/enhancement problems such as super-resolution and denoising have a long history [1,2,3,4].”
>
> [1] Tsai, R. Multiframe image restoration and registration. Advance Computer Visual and Image Processing 1 (1984): 317-339.
> [2] Park S C, Park M K, Kang M G. Super-resolution image reconstruction: a technical overview. IEEE signal processing magazine, 2003, 20(3): 21-36.
> [3] Rudin L I, Osher S, Fatemi E. Nonlinear total variation based noise removal algorithms. Physica D: nonlinear phenomena, 1992, 60(1-4): 259-268.
> [4] Candès E J, Romberg J, Tao T. Robust uncertainty principles: Exact signal reconstruction from highly incomplete frequency information. IEEE Transactions on information theory, 2006, 52(2): 489-509.
>
> 2. Thanks for your suggestion on the naming of the “Transformer”. We are also considering renaming it possibly into “transforming model”. But to keep the naming consistent throughout the discussion period, we will keep the original name for now.
>
> Thank you for your review again! Any further questions or suggestions are welcome.

---

### Official Review · AnonReviewer2 · 2019-10-19
**Official Blind Review #2**

**Rating:** 1

**Review:**

This paper presents several models for visual recognition in the presence of image degradation (e.g., low-resolution, noise, compression artifacts). In the models, an image enhancement network is placed in front of a recognition model and trained together with the recognizer to improve the recognition accuracy as well as to enhance the image quality. The proposed approach is simple, straightforward, yet effective. It has been also shown that the image enhancement module is transferable between different recognition tasks and architectures.

Although the paper addresses a timely topic and the performance gain is substantial, my current decision is reject mainly because of its weakness in technical novelty and contribution. The proposed models are simple and straightforward combinations of two separate networks, one for image enhancement and the other for recognition. This approach also makes the entire networks overly heavy, and introduces hyper-parameters (e.g., lambda) that have to be carefully tuned. Overall, it was hard to find interesting ideas that future readers may learn from the paper.

Other comments:

The 2nd model based on knowledge distillation (KD) is called "unsupervised", which however sounds weird. As already mentioned in the manuscript, the teacher network for KD is trained in a fully supervised manner for the target task, so it cannot be considered as an unsupervised model. Further, the advantage of the 2nd model is marginal in practice.

The advantage of the transformer in the 3rd model is not clearly discussed. It is unknown in the paper why the 3rd model with the transformer works best in the experiments. Also, regarding the main goal of the paper (i.e., image enhancement not for human but for recognition networks), the reason for adopting the transformer is hard to understand.

The degrees of image corruption (e.g., down-sampling, noise, compression) applied during testing are not mentioned at all, although they are important to understand the empirical advantage of the proposed models.

The transferability is one of the most important benefit of the proposed model, but not convincing sufficiently. The proposed models are transferable between different object categories, but the plain models seem to be also transferable, sometime more transparently. Also, it is not clearly discussed what makes the proposed models attaining the transferability.

It would be nice to apply the proposed models to the ImageNet-C benchmark.

Missing references
- Studying Very Low Resolution Recognition Using Deep Networks, CVPR 2016
-  Benchmarking Neural Network Robustness to Common Corruptions and Perturbations, ICLR 2019

**Experience Assessment:**

I have read many papers in this area.

**Review Assessment: Checking Correctness Of Derivations And Theory:**

I assessed the sensibility of the derivations and theory.

**Review Assessment: Checking Correctness Of Experiments:**

I assessed the sensibility of the experiments.

**Review Assessment: Thoroughness In Paper Reading:**

I read the paper thoroughly.

---

> ### Author Response · Authors · 2019-11-10
> **Response to AnonReviewer2 [4/4]**
>
> In the table below, we examine the transferability of RA Processing between recognition architectures, using the same two tasks "snow" and "speckle_noise", with corruption level 5. Note the recognition loss used during training is from a ResNet-18, and we evaluate the improvement over plain processing on ResNet-50/101, DenseNet-121 and VGG-16. We observe that the improvement over plain processing is transferable among different architectures.
> ------------------------------------------------------------------------------------------------------------------
> Corruption Type                     Snow                                                   Speckle noise
> ------------------------------------------------------------------------------------------------------------------
> Evaluation on          R18   R50  R101   D121  V16              R18   R50   R101   D121   V16
> ------------------------------------------------------------------------------------------------------------------
> No Processing        10.7  16.6   20.9  21.7   10.5                7.7   11.7   14.5     18.6    7.1
> Plain Processing     34.5  39.1   44.6  41.1   27.4               36.6  42.4   47.7    43.0   31.3
> ------------------------------------------------------------------------------------------------------------------
> RA w/ R18                43.7  47.9   51.7  47.9   37.4               45.2  50.3   53.3    49.1   39.0
> ------------------------------------------------------------------------------------------------------------------
>
>
> The ImageNet-C results have also been included in Appendix-D for reference in the revision.
>
> 6. Missing reference.
> Thanks for pointing us to the related works. We have added the missing references in the related work section.
>
> Thank you again for your detailed review! We hope our answers address your concerns. If you have any further questions, we are very happy to answer.
>
> [1] Colorful Image Colorization. Zhang et al. ECCV 2016.
> [2] EnhanceNet:  Single Image Super-resolution through Automated Texture Synthesis. Sajjadi et al. ICCV 2017.
> [3] Delving into Transferable Adversarial Examples and Black-box Attacks. Liu et al. ICLR 2017.
> [4] The Space of Transferable Adversarial Examples. Tramèr et al. 2017.

---

> ### Author Response · Authors · 2019-11-10
> **Response to AnonReviewer2 [3/4]**
>
> 5. ImageNet-C benchmark.
> Thanks for the suggestion. ImageNet-C is indeed a related benchmark to our work. This benchmark focuses on improving the robustness of the recognition model under various image conditions, while our work focuses on making the images more recognizable by a conventional recognition model. Thus we believe the emphasis is slightly different.
>
> Despite having a different emphasis, ImageNet-C is still a dataset where we can evaluate our methods on. We have run some experiments on ImageNet-C to showcase the results. Since only the corrupted validation set but not the training set is provided by the authors, we divide the val set into two halves and use the first/second half for training/testing.
>
> In the table below, we evaluate RA Processing on all 17 types of corruptions, with corruption level 5. We observe that RA Processing brings consistent improvement over plain processing, sometimes by an even larger margin than the tasks considered in section 4.
> ----------------------------------------------------------------------------------------------------------------------------------------------------------------
> Type            orig  brit  contr defoc elast gaub  gaun glass impul jpeg  motn pixel  shot   satr   snow  spat speck zoom
> ----------------------------------------------------------------------------------------------------------------------------------------------------------------
> No Proc.     69.9  51.3   3.3     11.3   17.1    9.3     1.2    8.7     1.0      29.4   11.1   23.1    1.8    39.5   10.7    19.1    7.7     17.6
> Plain Proc.  N/A  59.9  18.3    25.3   18.9   21.5   21.8  20.1   24.1    43.0   42.4   50.1   24.9   54.4   34.5    60.8   36.6    17.0
> ----------------------------------------------------------------------------------------------------------------------------------------------------------------
> RA Proc.      N/A  61.4   30.7   33.8   35.4   27.0   32.8  25.3   35.1    46.1   48.2   54.0   35.2   57.1   43.7    63.0   45.2    31.9
> ----------------------------------------------------------------------------------------------------------------------------------------------------------------
>
> In the table below, we experiment with different levels of corruptions with corruption type "snow" and "speckle noise". We also evaluate our variants -- Unsupervised RA and RA with Transformer. We observe that when the corruption level is higher, our methods tend to bring more recognition accuracy gain.
>
> ----------------------------------------------------------------------------------------------------------
> Corruption Type                          Snow                                        Speckle noise
> ----------------------------------------------------------------------------------------------------------
> Corruption  Level          1      2         3       4      5                 1      2        3        4       5
> No Processing            46.7  23.6  28.0  17.6  10.7           50.5  42.8  22.9  14.5   7.7
> Plain Processing        57.1  45.1  46.0  37.1  34.5           60.3  57.0  48.4  43.2  36.6
> -----------------------------------------------------------------------------------------------------------
> RA Processing            60.3  51.7  51.7  45.7  43.7           62.7  60.8  54.2  50.3  45.2
> Unsupervised RA       60.2  51.3  50.6  43.6  41.5           62.9  60.5  53.8  49.4  43.9
> RA w/ Transformer    55.7  46.7  48.1  42.7  40.9           59.0  57.7  52.2  49.2  44.7
> -----------------------------------------------------------------------------------------------------------

---

> ### Author Response · Authors · 2019-11-10
> **Response to AnonReviewer2 [2/4]**
>
>
> (..continued) We mentioned in section 3.3 that here “unsupervised” is only in terms of training the image processing model P, but not the recognition model R. Our problem setting (section 3.1) assumes that the recognition model R is a given fixed pretrained model, and it can be pretrained either in full supervision or in an unsupervised manner, because its training is not part of our training process and we only use it as a loss function. We only concern about the training of the image processing model P, and the method described in section 3.3 allows us to train P without ground truth label of images. Also, the dataset used to train P is not necessarily the same dataset used to train R (as in section 4.4 & appendix B.2, when we evaluate transferability among recognition tasks, one model is trained with PASCAL VOC and the other model is trained with ImageNet), so even if R needs to be trained by us with full supervision of its dataset, P can still be trained with another dataset of interest without label supervision, so this method could sometimes be practically useful. We’ve also added more explanation on this point in the section 3.3 of our revision. We welcome suggestions on how to describe the method in a more clear manner.
>
> 2. “The advantage of the transformer in the 3rd model is not clearly discussed. It is unknown in the paper why the 3rd model with the transformer works best in the experiments. Also, regarding the main goal of the paper (i.e., image enhancement not for human but for recognition networks), the reason for adopting the transformer is hard to understand.”
>
> The transformer often performs best possibly because with this extra network in the middle, the capacity of the whole system is increased: in RA Processing the processing model P optimizes both processing and recognition loss, but now P optimizes processing loss while T optimizes recognition loss. We’ve added this in the result analysis at the end of section 4.1.
>
> The advantages and disadvantages of using this transformer model was discussed at the end of section 3.4. The reason for adopting this method might be not wanting to affect the original image processing performance (in terms of P’s outputs), or sometimes the better recognition performance as shown in experiments.
>
> 3. Degree of corruption
>
> The degree of image corruption (along with other settings) was mentioned at Appendix A as “Experimental Details”, due to space limit. To obtain the input images, for super-resolution, we use a downsampling scale factor of 4; for denoising, we add Gaussian noise on the images with a standard deviation of 0.1 to obtain the noisy images; for JPEG deblocking, a quality factor of 10 is used to compress the image to JPEG format. We have moved it to the experiment section in the main text to make it more clear.
>
> 4. “The transferability is one of the most important benefits of the proposed model, but not convincing sufficiently. The proposed models are transferable between different object categories, but the plain models seem to be also transferable, sometime more transparently. Also, it is not clearly discussed what makes the proposed models attaining the transferability.”
>
> Our “transferability” means the improvement over plain processing is transferable, not over “no processing”. Our baseline in transferability experiments is “plain processing”, but not “no processing”. The fact that plain processing’s improvement over no processing is not specific to any recognition model, is not that surprising in our opinion. This is because plain processing improves image’s overall quality, and it does not use a recognition model in training. The improvement on recognition accuracy of plain processing was shown in some prior works [1,2] which use recognition performance as the metric for processing quality, but is not the focus of our work.
>
> In this work, our aim is better recognition performance than plain processing (only using the traditional image processing loss as in most prior works), and in that regard, the improvement is shown to be transferable in multiple conditions (architectures, categories, recognition tasks) in experiments.
>
> One of the reasons why our method attains transferability is possibly that these models learn many common features that could be useful for general computer vision, especially in shallower layers. More importantly, the reason could be similar to the reason why adversarial examples can transfer among models: different models’ decision boundaries are similar. [3] studies adversarial exmaples’ transferability and shows decision boundaries of different models align well with each other; [4] quantitatively analyzes similarity of different models' decision boundaries, and shows that the boundaries are close in arbitrary directions, whether adversarial or benign. We added this discussion in section 4.2. Thanks for your question.

---

> ### Author Response · Authors · 2019-11-10
> **Response to AnonReviewer2 [1/4]**
>
> Thank you for your constructive feedback! We are happy to address your concerns below and we have uploaded a revision reflecting the changes. For easier reading, we’ve pasted some of your comments in our response, and please bear with the length of our response.
>
> Response to major concern:
>
> Technical novelty and contribution
> We are glad to see the reviewer agrees that we are addressing an important and timely problem (making image processing outputs more accurately recognized by machines), and our method brings substantial performance gain. We agree that our method towards this goal is simple and straightforward, but we would like to view this simplicity as a strength. It makes our methods easy to implement and potentially more widely used in practice. Our contribution does not lie in designing network architectures or components, but is to showcase our simple methods can work favorably on an important but largely ignored problem, and more interestingly the improvement is even “transferable”. We agree that architecture-wise, our method consists of two separate networks, but the key idea is to use a fixed recognition loss upon the original image processing loss for better machine recognizability of image processing outputs. Our technical contributions are bringing this problem to the community, developing a simple and effective method which imposes a recognition loss, with its variants that are useful in different scenarios, and showing the performance gain is transferable under various conditions.
>
> “Overly heavy” system
> Our system might be “heavier” during training than plain processing, since it incorporates an additional loss computation with the recognition model. Our training still finishes in a reasonable amount of time (less than one hour to a few hours with a single GPU). More importantly, once the training is finished, the recognition model used as loss is not needed anymore, and during inference, we only need the processing model P, so no additional overhead is introduced when the model is actually put to deployment. We have included this point in section 3.2 in the revision.
>
> Hyperparameter $\lambda$
> Incorporating a new loss function often requires tuning of the coefficient hyperparameter, but in our case this hyperparameter $\lambda$ is only grid searched once within a short range for each variant of our methods (RA Processing, Unsupervised RA and RA w/ Transformer), using ResNet-18 as the recognition model and super-resolution as processing tasks. The same $\lambda$ is then used on other processing tasks and recognition models. This means a consistent $\lambda$ works well with different conditions. An analysis of the hyperparameter $\lambda$ with RA Processing is presented at Table 6, we can see that $\lambda$ from 1e-4 to 1e-2 all bring substantial improvement in terms of recognition accuracy. As for each variant of our method, this brief grid search is necessary since unsupervised RA and RA processing have very different forms of loss functions and the two losses differ a lot in scales.
>
> “It was hard hard to find interesting ideas that future readers may learn from the paper”
> Overall, we raised an important problem to the community, developed a simple method (and several variants for different use cases) that can work on the problem, and presented the intriguing “transferability” of our method. This transferability phenomenon is quite surprising to us. It could bring insights into questions like how neural networks function and what do different neural networks share in common. We believe our work can be useful to the research community as a new problem is raised, and we hope it encourages researchers to further develop better methods on this problem. Also our method could be useful for industry usage as it is simple and gives substantial performance gain on a very practical problem.
>
>
> Response to other comments:
> 1.“The 2nd model based on knowledge distillation (KD) is called "unsupervised", which however sounds weird. As already mentioned in the manuscript, the teacher network for KD is trained in a fully supervised manner for the target task, so it cannot be considered as an unsupervised model. Further, the advantage of the 2nd model is marginal in practice.”
>
> First we would like to clarify on the relation with KD. Our system is similar to the KD in terms of the loss function (using a predicted “soft” probability distribution to guide the training instead of “hard” ground truth labels), but not in terms of the “teacher-student” model paradigm. In our system, the probability distribution is obtained by feeding the original image to the same pretrained recognition model, but in KD, it’s obtained by feeding the image to a different teacher model. Thus, the pretrained model R is not considered as a “teacher model”, but the original image can be considered as a “teacher image”.

---

### Official Review · AnonReviewer1 · 2019-10-24
**Official Blind Review #1**

**Rating:** 3

**Review:**

The goal in this work is to improve machine interpretability of images.
The authors main claims are:
-	Their proposed approach improves image recognition accuracy even without knowing subsequent recognition tasks and recognition models used to perform them (transferable model to different recognition models/tasks).
-	For this they propose what they call “Recognition-Aware” processing that combines image processing loss and recognition loss.
-	The approach is evaluated on three image processing tasks with two downstream recognition tasks:
o	Image super-resolution, de-noising, and JPEG-de-blocking processing tasks, with
o	Image classification and object detection recognition tasks.

The paper is well written and organized, experiments carried are extensive but the reuse of known neural networks, many simplifications (shortcuts), a not clear enough methodology (see below), limited processing & recognition tasks used to support it, do not justify in our opinion the main (over-arching) work’s claim:
-	In 3.2 optimizing recognition loss/Last paragraph:  “Interestingly, we find that image processing models trained with the loss of one recognition model R1, can also boost the performance when evaluated using recognition model R2, even if model R2 has a different architecture, recognizes a different set of categories or even is trained for a different task.”.

The paper would greatly benefit (to understand the context of the work or the explanations provided) from clarification of the many under-defined, not clearly introduced concepts it carries:
-	Meaning of “Network” is not clearly defined:
o	Abstract: “image processing network”.
o	Introduction: “the network maps an image to a semantic label”
o	Later in the paper only networks introduced are deep neural networks. That should be clear from beginning of the paper.
-	“Retraining/Adaptation”  in 1st paragraph page 2.
-	In 1. Introduction/Paragraph 1: You use “.. techniques .. have been proposed for making the output images look natural to human”:
o	Noise is part of nature. A de-noised (smoothed) image is not “more natural”.
o	Enhanced (processed) images are not necessarily “more” natural, rather they take advantage of the human visual perception characteristics to enhance recognition for example.
-	In 1. Introduction/Paragraph 3:
o	“.. of great importance that the processed images be recognizable”  Should explain the concept of image recognition! Because it could be related to contained objects, overall description (for captioning for example) etc.
-	“Image processing” in the context of the paper is intended only as “image enhancement for recognition”. Pattern detection, segmentation, object extraction etc. are not included in this restrictive definition. Should specify for example: image enhancement and restoration.
-	Figure 1: As an illustration, it’s completely counterproductive for your discourse as many simple image recognition algorithms would recognize the bird even in the noisy image.
-	In 3. Unsupervised optimization of recognition loss: The “unsupervised RA” process is not  clear enough to us especially the statement:
o	“.. only “unsupervised” for training model P, but the target pre-trained model R can still be trained in full supervision.”.


-	“We may not know what network architectures (e.g. ResNet or VGG) will be used for inference, what object categories the downstream model recognizes (e.g. animals or scenes), or even what task will be performed on the processed image (e.g. classification or detection)”.
o	Is your goal a universal “recognition model” applicable to anything?
-	I also have some trouble with the terminology:
o	In 1. Introduction/Paragraph 4: “It is also important that the enhanced machine semantics is not specific to any concrete recognition model”: “enhanced machine semantics”!
o	In 1. Introduction/Paragraph 4: “..transferable among different recognition architectures..”. Does “architectures” refer to deep neural networks (DNN)? If yes, is recognition performed only by DNN? What about the preceding bullet (“is not specific to any concrete recognition model”)?
-	In 1. Introduction/Paragraph 3:
o	 “.. we argue that image processing systems should maintain/enhance machine semantics”. Do not see what’s to argue here?
o	“Recognition-Aware Image Processing” is it simply put Image Processing techniques for recognition enhancement (“Recognition” still needs to be defined)?
-	In 2 Related work :
o	 “ .. we assume we do not have the control on the recognition model, as it might be on the cloud or decided in the future, thus we advocate adapting the image processing model only. This also ensures the recognition model is not harmed on natural images.” Care to explain?
o	: “to achieve better recover the face identity from low-resolution images”, Typo?
-	In 1. Introduction/Paragraph 1:
o	“ .. might not look “natural” to machines”: Care to explain this concept?
	Would advise to just keep the second part of the sentence.
-	In 1. Introduction/Paragraph 2: “One could specifically train a recognition model only on these output images produced by the de-noising model to achieve better performance on such images, but the performance on natural images can be harmed.”  Care to explain?.
o	More complicated images (noisier, multiple obstructions etc.) are recognized nowadays and true to actual applications.

-	3.4 using an intermediate transformer/Last paragraph:
o	“ .. that there are two instances for each image (the output of model P and T), one is “for human” and the other is “for machines”.”:
	The “Transformer” characteristics are not clearly defined for the intended output (For machines?).
	Why is output of model T not represented in Figure 2 (Right)?

**Experience Assessment:**

I have read many papers in this area.

**Review Assessment: Checking Correctness Of Derivations And Theory:**

I assessed the sensibility of the derivations and theory.

**Review Assessment: Checking Correctness Of Experiments:**

I assessed the sensibility of the experiments.

**Review Assessment: Thoroughness In Paper Reading:**

I read the paper at least twice and used my best judgement in assessing the paper.

---

> ### Author Response · Authors · 2019-11-09
> **Response to AnonReviewer1 [3/3]**
>
> Fourth paragraph:
> 1. About the Transformer model.
> a) Transformer characteristics
> Here the transformer T takes an input image from the image processing model P, and output an image that is optimized for machine recognition, thus creating the “two instances of images” situation. With the help of T, the processing model P focuses on optimizing the processing loss and T focuses on optimizing the recognition loss (Eqn. 6). Because the recognition loss is only imposed on the output of T, and the gradient is cut off from flowing back to P, it is as if there’s no recognition loss to P. Thus the output of P (input of T) is guaranteed not affected as for human perception.
>
> In the last paragraph of section 3.4 we discussed the pros and cons of using this Transformer model instead of using the most simple variant of our method (RA Processing): it can guarantee performance for human perception in terms of output from P, but also create this “two-image” situation. In practice one can choose whether to use the Transformer based on practical needs.
>
> b) Figure 2 (right).
> This is mainly due to the space/page width limit and the “recognition loss” part is the same as Figure 2 left (dashed box, “Recognition Loss”), so we use this to save some space. We are happy to include the full figure for clarity if needed.
>
> Thank you again for your detailed review! We hope our response addresses your concerns. Any further questions or suggestions are welcome.
>
> [1] Classification-driven dynamic image enhancement. Sharma et al. 2018.
> [2] Task-driven super resolution: Object detection in low-resolution images. Haris et al. 2018.
> [3] Benchmarking neural network robustness to common corruptions and perturbations. Hendrycks et al. 2019.
> [4] Episodic training for domain generalization. Li et al. 2019.
> [5] Generalizing across domains via cross-gradient training. Shankar et al. 2018.

---

> ### Author Response · Authors · 2019-11-09
> **Response to AnonReviewer1 [2/3]**
>
> Third paragraph:
> 1. Is your goal a universal “recognition model” applicable to anything?
> Our goal is to learn an image processing model so that the  processed  images can be more accurately recognized by various downstream recognition models, rather than a processing model that just aims at good-looking images to human as most previous works focus on.
>
> 2. Terminology
> a) Here “enhanced machine semantics” has the same meaning as “better machine recognizability”, which means images more accurately recognized by machines. We have changed it to “enhanced machine recognizability” to make it more clear.
> b) Yes, it means different neural network architectures, and in this work the recognition is only performed by DNN. We have changed “architectures” into “neural network architectures” in the revision. We agree that “is not specific to any concrete recognition model” is a too broad description and have changed it to “is not specific to any concrete neural network-based recognition model”, but we also would like to point out that neural networks are currently popular choices as image processing/recognition models, and are more related to the ICLR community.
>
> 3. Intro paragraph 3.
> a) Thanks for asking. Here a more accurate expression than “argue” would be “advocate”, and we have changed the word. Also we changed “machine semantics” to “machine recognizability”. Prior works didn’t explicitly try to optimize for “machine semantics/recognizability” and in this paper we “advocate” it’s also important, other than human perception.
>
> b) This is a good summary of our work but we would also like to add that our work is not trying to optimize recognition accuracy by processing images, it’s more about making image processing outputs better recognized by recognition models. Traditionally, people try to make the image processing outputs look good to human, but we develop techniques to make them also better recognized by machines. Here recognition is defined as identifying the content/object of the image (e.g., classification, object detection).
>
> 4. Related work
> Explanation of: “ .. we assume we do not have the control on the recognition model, as it might be on the cloud or decided in the future, thus we advocate adapting the image processing model only. This also ensures the recognition model is not harmed on natural images.”
>
> In past works, when a recognition model is used to guide an image processing model, it was shown that the recognition accuracy on that particular recognition model can be improved. But in this work, we show that we can develop approaches such that the improvement is transferable among different recognition models, as we explained in (#2) above. This transferability property removes the requirement that we must have access/control to the particular recognition model on which we want to improve accuracy, as we can resort to other models in training.
>
> Also, in past works, the recognition model is mostly fine-tuned together, but we do not fine-tune the recognition model jointly with the image processing model (this is what we mean by “adapting the image processing model only”) — the recognition model are pretrained and only serves as a loss function. Its weights are fixed, thus guaranteeing that its accuracy on normal/natural images would not degrade. We have shown in section 5 (“Fine-tuning the Recognition Model”) that if it is jointly trained, the accuracy on normal images would be hurt.
>
> “to achieve better recover the face identity from low-resolution images”
> Thanks for pointing out the typo, we have removed the word “achieve”.
>
> 5. “ .. might not look “natural” to machines”
> Following your advice we have removed “ .. might not look ‘natural’ to machines” and only kept the second part of the sentence.
>
> 6. a) Explanation of “One could specifically train a recognition model...”: please refer to our response #2 in the first paragraph of questions.
> b) “More complicated images (noisier, multiple obstructions etc.) are recognized nowadays and true to actual applications.”
> Yes, we agree that now some recognition systems can robustly recognize noisy/obstructed content, but many models are still trained on normal images and perform best when tested in such cases, where our methods could be used. Our work is different from those works which aim for robustness of the recognition model since we focus on training of the processing model and assume the recognition model is given. We have added related references [3,4,5] in the related work.

---

> ### Author Response · Authors · 2019-11-09
> **Response to AnonReviewer1 [1/3]**
>
> Thank you for your constructive feedback! We answer your questions below, and we have uploaded a revision addressing your concerns. For easier reading we pasted some of your comments, and please bear with the length of our response.
>
> First paragraph:
> “the reuse of known neural networks, many simplifications (shortcuts), a not clear enough methodology (see below), limited processing & recognition tasks used to support it, do not justify in our opinion the main (overarching) work’s claim”
>
> “reuse of known neural networks”
> We would like to clarify that our contribution does not lie in designing new neural architectures, but the use of recognition loss on image processing outputs. To demonstrate the general usefulness of our method, we chose popular neural networks (SRResNet, VGG, ResNet, DenseNet) in the literature for experiments.
>
> “many simplifications (shortcuts)”
> Could you elaborate more on this point? Sorry but we are not sure what “simplifications” refer to here. If “simplifications” refers to the use of neural networks as processing/recognition models, we acknowledge this point, but we also would like to mention that NNs are currently popular models for such tasks.
>
> “a not clear enough methodology (see below)”
> We answer your detailed questions about our method below.
>
> “limited processing & recognition tasks”
> We experimented with three image processing tasks & two recognition tasks (in total six pairs), and transferability between the two recognition tasks, for all three of our main methods and five recognition architectures, thus we respectfully disagree that our used tasks are limited. Most prior works (e.g., [1,2]) only consider one processing/recognition task. In the revision, we also added some results on the ImageNet-C benchmark [3] which has 17 types of corruptions in Appendix D.
>
> “do not justify in our opinion the main (overarching) work’s claim”
> We believe the experiments in section 4.2-4.4 and appendix B demonstrated how the performance gain is transferable (as in the claim) under these various conditions.
>
> Second paragraph:
> 1. Definition of “Network”
> The “network” mentioned in the paper means a (deep) convolutional neural network, we have clarified this in abstract in the revision (network -> neural network), following your suggestion.
>
> 2. “Retraining/Adaptation”
> Here “retraining/adapting” means training a recognition specifically on the image processing outputs (instead of natural images as usual) or adapting (e.g., using some domain adaptation approaches) the naturally trained image recognition model so that it specifically recognizes images output by an image processing model (e.g., denoised images). This was briefly explained in the sentence before, and we’ve made it more clear in the revision.
>
> 3. “look ‘natural’ to human”
> We agree that denoised or enhanced images are not necessarily more “natural”, and we’ve changed it to “for making the output images more perceptually pleasing to human”. Thanks for your suggestion.
>
> 4. Definition of recognition.
> In the revision, we added a brief explanation about “recognizable” in paragraph 3: “In other words, recognition systems (e.g., image classifier or object detector), should be able to accurately explain the underlying semantic meaning of the image content .” In our context, this recognition system could be any neural network including a neural-based captioning system.
>
> 5. Specifying as enhancement/restoration.
> Following your suggestion we have changed in the last paragraph of introduction from “We conduct extensive experiments, on multiple image processing tasks” to “We conduct extensive experiments, on multiple image enhancement/restoration tasks”. We also added a clarification in the experiment section: “More specifically, these are image enhancement or restoration tasks, where usually the target image is an enhanced image or the original image. Other more broader image processing tasks such as pattern detection, segmentation, object extraction are not considered in this work.” We will also consider updating the title to reflect this modification.
>
> 6. Figure 1.
> We agree that many recognition systems would still recognize this particular noisy bird image correctly, but the accuracy on noisy images is also severely hurt on noisy images compared with normal ones. In this case, we used a modern network architecture (i.e., ResNet-18) and it indeed incorrectly classifies this noisy image as a kite.
>
> 7. Meaning of “unsupervised”.
> This “unsupervised RA” scheme means in the recognition loss we regress the probability output of the original image (“soft label”), rather than the hard label in the supervised case. “only ‘unsupervised’ for training model P” means that in training the image processing model P, we do not need image labels, but the recognition model R could be trained in any manner, either with or without full label supervision, because we assume R is a pretrained model in our problem setting. We’ve added more clarification in the revision.

---

### Decision · Program_Chairs · 2019-12-19

**Decision:**

Reject

**Comment:**

This paper presents several models for recognition-aware image enhancement. The authors propose to enhance the image quality in the presence of image degradation (e.g., low-resolution, noise, compression artifacts) as well as to improve the recognition accuracy in a joint model. While acknowledging that the paper is addressing an interesting direction, the reviewers and AC note the following potential weaknesses: presentation clarity, limited technical contributions, insufficient empirical evidence. AC can confirm all the reviewers have read the rebuttal and have contributed to the discussion. All the reviewers and AC agree that the rebuttal was informative, and the authors have partially addressed some of the concerns (e.g. additional experiments). R2 has raised the score from reject to weak reject. However, at this stage AC suggest the manuscript is below the acceptance bar and needs a major revision before submitting for another round of reviews. We hope the reviews are useful for improving and revising the paper.